# OpenReview forum: "Cognitive Structure Generation via Diffusion Models with Policy Optimization"
_ICLR.cc/2026/Conference — Submitted to ICLR 2026_

### Official Review · Reviewer_Aje3 · 2025-10-26

**Soundness:** 3
**Presentation:** 3
**Contribution:** 2
**Rating:** 6
**Confidence:** 4

**Summary:**

This work frames student modeling as Cognitive Structure Generation (CSG): given a student’s learning history, generate a personalized graph. The nodes encode concept mastery and edges encode inter-concept relations. It instantiates this with a Cognitive Structure Diffusion Probabilistic Model (CSDPM) trained in two stages: (i) pretraining on simulated cognitive structures inferred from logs via a rule-based procedure, and (ii) policy optimization of the reverse diffusion process to align generated graphs.

**Strengths:**

- Overall, I like the idea and the problem setting behind the proposed methods. Modeling dynamic, inter-concept graphs is both important and challenging,  especially given the limited data available per student, the lack of concrete evaluation methods, and the difficulty of optimizing over discrete structures. I also appreciate the effort to combine KT and CD, since prediction and interpretability are equally important in the educational domain.

- The method’s use of a discrete graph diffusion model combined with policy optimization guided by the SOLO taxonomy is interesting. The pipeline, i.e., pretraining on simulated structures followed by reinforcement learning (RL) alignment, is novel within the education modeling space.

- The paper is very well-presented. I enjoyed both the clarity of the writing and the quality of the figures.

**Weaknesses:**

- The training signal for Stage-I comes from rule-based simulation computed from the same interaction logs used downstream, with Gaussian noise added t the Q-matrix for robustness. The mapping $f_{\mathrm{UOC}}, f_{\mathrm{UOR}}$ is basically weighted correctness; then values are rounded and one-hot encoded, which discards uncertainty and may bake in label-like information before KT/CD. A stronger justification or comparison against alternative simulators (Bayesian knowledge tracing, IRT-based posteriors, or human-elicited maps) is needed.
- The 8:1:1 split at the interaction level seems somewhat inconsistent with the paper’s introduction section about “generalization” and “cold-start”. It would be informative to see how the models perform under smaller data regimes, especially given that several heuristics are already used so the method should work with small data.
- The edge construction $f_{\text {UOR }}$ uses co-occurrence within items as evidence of a relation between concepts. This conflates test design with student cognition; edge identifiability is questionable. Visual cases are interesting but anecdotal. A human-judged metric (or even prompt LLMs to evaluate edges) would make it more concrete.
- Three datasets have only 20 items, and the largest uses 57 concepts but task heterogeneity is unclear. It’s hard to conclude that CSG scales to hundreds–thousands of concepts or to more realistic, noisy Q-matrices. Reported generation times are low, but training cost of diffusion + RL may be substantial.
- The graph size is fixed (predefined by the number of concepts). In real-world educational settings, graph structures are dynamic: concept sets can expand or contract, and nodes often exist at different abstraction levels. For example, “linear algebra” as a field, “dot product” as a concept, and “specific exercises” as instances, which I think are all presented in student's mind. How does this method extend to these situations?

**Questions:**

See weaknesses.

---

> ### Author Response · Authors · 2025-11-21
> **Response to Reviewer Aje3 (Part 1/4)**
>
> We sincerely appreciate your encouraging reviews, which have been extremely helpful in our future work. We address individual points below.
>
>
> > The training signal for Stage-I comes from rule-based simulation computed from the same interaction logs used downstream, with Gaussian noise added t the Q-matrix for robustness. The mapping f_UOC, f_UOR is basically weighted correctness; then values are rounded and one-hot encoded, which discards uncertainty and may bake in label-like information before KT/CD. A stronger justification or comparison against alternative simulators (Bayesian knowledge tracing, IRT-based posteriors, or human-elicited maps) is needed.
>
>
> Thank you for this detailed comment. We agree with the reviewer that f_UOC, f_UOR can be viewed as forms of weighted correctness computed directly from interaction logs. Stage I is designed to provide a lightweight, model-free proxy for cognitive structures when no ground-truth graphs are available.
>
>
> Alternative simulators are certainly possible but come with notable trade-offs:
> - BKT-/IRT-based posteriors require fitting separate models to response data, introducing additional training overhead, hyperparameter tuning, and potential model-mismatch issues.
> - Human-elicited cognitive maps provide valuable information but are costly to obtain at scale and are unavailable for most educational datasets.
>
>
> Given these constraints, the simple rule-based proxies offer a practical and domain-agnostic initialization for Stage I. Stage I pretraining is also followed by SOLO-guided reinforcement learning in Stage II, which allows the generator to move beyond any biases in the rule-based initialization. Our ablation results show that Stage II significantly refines the structures, indicating that the model does not simply inherit the handcrafted rules but learns more accurate patterns from data.
>
>
> We also ensure that there is no label leakage for our KT and CD downstream tasks. Stage I simulation and pretraining use only the training split; no test interactions are ever used to construct simulated structures or to train CSG. We have also incorporated the discussion above into our paper revision.
>
>
> > The 8:1:1 split at the interaction level seems somewhat inconsistent with the paper’s introduction section about “generalization” and “cold-start”. It would be informative to see how the models perform under smaller data regimes, especially given that several heuristics are already used so the method should work with small data.
>
>
> Thank you for this thoughtful comment. We have conducted an additional experiment in which we reduce the downstream training set for CSG by 50% while keeping the baselines trained on the full dataset. The results show that CSG continues to outperform the best-performing baselines on both KT and CD even with only half of the training data. This suggests that the representations learned by CSG transfer effectively and remain robust under more limited data conditions.
>
>
> |Task|Dataset|Math1|||Math2|||FrcSub|||NIPS|||
> |---|---|---|---|---|---|---|---|---|---|---|---|---|---|
> ||Metric|AUC↑|ACC↑|RMSE↓|AUC↑|ACC↑|RMSE↓|AUC↑|ACC↑|RMSE↓|AUC↑|ACC↑|RMSE↓|
> |KT|PSI-KT|0.8118|0.7392|0.4317|0.7759|0.7140|0.4403|0.8533|0.7908|0.3309|0.7260|0.6687|0.4520|
> ||CSG-KT|0.8220|0.7412|0.4283|0.7772|0.7197|0.4390|0.8636|0.8022|0.3192|0.7413|0.6757|0.4511|
> ||CSG-KT-50%|0.8139|0.7328|0.4314|0.7702|0.7159|0.4070|0.8544|0.7970|0.3276|0.7382|0.6725|0.4547|
> |CD|DisenGCD|0.7983|0.7628|0.4001|0.8039|0.7457|0.4324|0.8559|0.8375|0.3342|0.7886|0.7311|0.4275|
> ||CSG-CD|0.8133|0.7710|0.3987|0.8179|0.7521|0.4270|0.8699|0.8451|0.3152|0.8036|0.7507|0.4242|
> ||CSG-CD-50%|0.8094|0.7646|0.4038|0.8109|0.7454|0.4309|0.8602|0.8373|0.3190|0.7987|0.7479|0.4241|

---

> > ### Author Response · Authors · 2025-11-21
> > **Response to Reviewer Aje3 (Part 2/4)**
> >
> > > The edge construction f_UOR uses co-occurrence within items as evidence of a relation between concepts. This conflates test design with student cognition; edge identifiability is questionable. Visual cases are interesting but anecdotal. A human-judged metric (or even prompt LLMs to evaluate edges) would make it more concrete.
> >
> >
> > Sorry for the confusion and we clarify f_UOR does not rely solely on item-level co-occurrence in the Q-matrix. Instead, it combines co-occurrence with weighted correctness on items where concepts $k_a$ and $k_b$ appear together. As a result, co-occurrence alone does not automatically produce a relation: if a student consistently struggles on these joint items, the relation score stays low; only when the student shows consistent success on such items is the relation treated as constructed.
> >
> >
> > Following the reviewer’s suggestion, we have conducted an interpretability study by comparing model-generated cognitive structures with human expert annotations. We asked annotators with MS and PhD backgrounds in education to label the cognitive-structure graphs of 25 randomly selected students across 20 interactions each (500 graphs) for both the FrcSub and Math2 datasets. These annotations serve as a proxy for ground-truth cognitive structures. For LLM-generated structures, we prompted GPT-5 and Llama-3-70B to produce binary adjacency matrices. We then evaluated both our CSG outputs and the LLM-generated graphs using Jaccard similarity and Graph Edit Distance (GED) against the human annotations. The results (tabulated below) show that CSG consistently achieves higher Jaccard similarity and lower GED, indicating that the cognitive structures it produces are more faithful to human-interpretable patterns. This provides quantitative evidence supporting the interpretability of our method.
> >
> >
> > |Datasets|Proxy & Generator|Jaccard↑|Graph Edit Distance↓|
> > |---|---|---|---|
> > |FrcSub|GPT-5|0.65|0.30|
> > ||Llama-3-70B|0.39|0.61|
> > ||CSG|**0.79**|**0.15**|
> > |Math2|GPT-5|0.54|0.43|
> > ||Llama-3-70B|0.34|0.67|
> > ||CSG|**0.69**|**0.21**|
> >
> >
> > To further assess the quality of the learned cognitive structures, we also compare CSG with LLM-generated structures on downstream KT and CD tasks. For each of the FrcSub and Math2 datasets, we randomly sampled 500 students and asked GPT-5 and Llama-3-70B to generate cognitive-structure graphs using the same annotation protocol as above. We trained KT and CD models on these graphs using identical optimization procedures for all three approaches. The performance on the 500-student subsets is summarized below. We find that CSG consistently outperforms both LLM-based baselines across all KT and CD metrics. The improvement is especially pronounced for KT, suggesting that CSG captures temporal evolution of cognitive structures more effectively. Although Llama-3-70B is substantially larger and strong on standard benchmarks, its generated cognitive structures do not translate into competitive KT/CD performance, whereas CSG provides more accurate and task-relevant modeling of student cognitive structures.
> >
> >
> > |Tasks|Datasets|FrcSub|||Math2|||
> > |---|---|---|---|---|---|---|---|
> > ||Metrics|AUC↑|ACC↑|RMSE↓|AUC↑|ACC↑|RMSE↓|
> > |KT|GPT-5-KT|0.7430|0.6683|0.3754|0.6785|0.6027|0.4840|
> > ||Llama-3-70B-KT|0.5170|0.4869|0.4139|0.4595|0.4030|0.5025|
> > ||CSG-KT|**0.8602**|**0.8010**|**0.3197**|**0.7457**|**0.6854**|**0.4283**|
> > |CD|GPT-5-CD|0.7858|0.7545|0.3462|0.7452|0.6759|0.4683|
> > ||Llama-3-70B-CD|0.5569|0.5376|0.3856|0.5063|0.4381|0.5077|
> > ||CSG-CD|**0.8691**|**0.8433**|**0.3168**|**0.7885**|**0.7247**|**0.4389**|

---

> > > ### Author Response · Authors · 2025-11-21
> > > **Response to Reviewer Aje3 (Part 3/4)**
> > >
> > > > Three datasets have only 20 items, and the largest uses 57 concepts but task heterogeneity is unclear. It’s hard to conclude that CSG scales to hundreds–thousands of concepts or to more realistic, noisy Q-matrices. Reported generation times are low, but training cost of diffusion + RL may be substantial.
> > >
> > >
> > > Thank you for this suggestion. We agree that examining performance on datasets with larger concept spaces is important. We have added experiments on the ASSISTments17 dataset, which contains 101 knowledge concepts and 2,210 questions. As shown in the tables below, our CSG-based models continue to yield consistent improvements over strong KT/CD baselines at this larger scale.
> > >
> > >
> > > |Task|Method|AUC↑|ACC↑|RMSE↓|
> > > |---|---|---|---|---|
> > > |KT|DKT|0.6832|0.6683|0.4543|
> > > ||SAKT|0.6762|0.6703|0.4601|
> > > ||GKT|0.7226|0.7092|0.4502|
> > > ||SKT|0.7349|0.7242|0.4457|
> > > ||GRKT|0.7493|0.7359|0.4426|
> > > ||MIKT|0.7677|0.7481|0.4403|
> > > ||ENAS-KT|0.7812|0.7627|0.4384|
> > > ||simple-KT|0.7791|0.7605|0.4389|
> > > ||PSI-KT|0.7890|0.7653|0.4337|
> > > ||CSG-KT|**0.7963**|**0.7792**|**0.4313**|
> > > |CD|IRT|0.7381|0.6847|0.4694|
> > > ||MIRT|0.7558|0.6950|0.4546|
> > > ||NCD|0.7687|0.7064|0.4462|
> > > ||RCD|0.7769|0.7157|0.4431|
> > > ||HyperCDM|0.7927|0.7301|0.4387|
> > > ||DisenGCD|0.7842|0.7249|0.4392|
> > > ||CSG-CD|**0.8003**|**0.7386**|**0.4320**|
> > >
> > >
> > > We also appreciate the reviewer’s note on scenarios with hundreds or thousands of concepts. When cognitive structures are required across many subjects or grade levels, a practical approach is to partition the full concept set into topical groups and apply CSG training and inference within each group. Exploring principled strategies for such hierarchical or modular training is a promising direction for future work.
> > >
> > >
> > > Regarding training cost, we acknowledge that diffusion models involve iterative denoising and can be computationally demanding. Recent advances in diffusion modeling provide general solutions that can directly accelerate our framework. In particular, latent-space diffusion models [1] substantially reduce training and sampling cost by operating in a compressed representation space, and improved noise-schedule design [2] has been shown to significantly enhance training efficiency. These techniques are complementary to our contribution and can be integrated into future versions of CSG to further reduce training time.
> > >
> > >
> > > Finally, once pretrained, the CSG framework amortizes the cost of graph structure learning across downstream KT and CD tasks. This leads to more efficient fine-tuning, reduced parameter count, and lower computational burden in deployment.
> > >
> > >
> > > > The graph size is fixed (predefined by the number of concepts). In real-world educational settings, graph structures are dynamic: concept sets can expand or contract, and nodes often exist at different abstraction levels. For example, “linear algebra” as a field, “dot product” as a concept, and “specific exercises” as instances, which I think are all presented in student's mind. How does this method extend to these situations?
> > >
> > >
> > > Thank you for raising this point. Our current work focuses on the setting commonly assumed in KT/CD [3–5] and in many deployed educational systems: a curriculum- or assessment-defined concept set (i.e., the Q-matrix ontology) that remains relatively stable within a course or semester [6]. Within this fixed ontology, CSG models the temporal evolution of a student’s cognitive structure without requiring additional fine-tuning.
> > >
> > >
> > > From the perspective of educational psychology [7] and topological psychology [8], a learner’s knowledge state is often characterized as an internally organized network of concepts and relations. Our generated cognitive structures are designed as an approximation of this representation: nodes correspond to constructed states of key concepts (e.g., “linear algebra,” “dot product”), and edges capture the relations that the learner has formed among them.
> > >
> > >
> > > We also appreciate the reviewer’s suggestion regarding different abstraction levels and evolving concept sets. The CSG framework is not inherently limited to a flat concept layer. In principle, it can be extended to multi-level or heterogeneous graphs, where nodes represent domains, intermediate concepts, or finer-grained skills, and edges describe relations both within and across levels. New concepts can be incorporated without retraining the entire system. For example, one could (i) pretrain the diffusion backbone on a broader ontology and fine-tune it when new concepts appear [9,10], or (ii) initialize embeddings for new concepts from textual or ontological neighbors [11] and continue diffusion+RL training with mild regularization to preserve existing structures. More generally, inductive mechanisms such as feature-based initialization, adapter layers, or continual-learning approaches [12] can be integrated to support dynamically expanding concept sets. We have added this discussion to the future work section (Appendix J) in the revised manuscript.

---

> > > > ### Author Response · Authors · 2025-11-21
> > > > **Response to Reviewer Aje3 (Part 4/4)**
> > > >
> > > > References
> > > >
> > > >
> > > > [1] Rombach R, Blattmann A, Lorenz D, et al. High-resolution image synthesis with latent diffusion models. CVPR 2022.
> > > >
> > > >
> > > > [2] Hang T, Gu S, Bao J, et al. Improved noise schedule for diffusion training. CVPR 2025.
> > > >
> > > >
> > > > [3] Shiwei Tong, Qi Liu, et al. Structure-based knowledge tracing: An influence propagation view. ICDM 2020.
> > > >
> > > >
> > > > [4] Weibo Gao, Qi Liu, et al. RCD: relation map driven cognitive diagnosis for intelligent education systems. SIGIR 2021.
> > > >
> > > >
> > > > [5] Shangshang Yang, Mingyang Chen, et al. Disengcd: A meta multigraph-assisted disentangled graph learning framework for cognitive diagnosis. NeurIPS 2024.
> > > >
> > > >
> > > > [6] Tyler R W. Basic principles of curriculum and instruction.Curriculum studies reader E2. Routledge, 2013.
> > > >
> > > >
> > > > [7] D. P. Ausubel, Educational psychology: A cognitive view, 1968.
> > > >
> > > >
> > > > [8] Kurt Lewin. Principles of topological psychology. Read Books Ltd, 2013.
> > > >
> > > >
> > > > [9] Ruiz N, Li Y, Jampani V, et al. Dreambooth: Fine tuning text-to-image diffusion models for subject-driven generation. CVPR 2023.
> > > >
> > > >
> > > > [10] Zhuang Z, Zhang Y, Wang X, et al. Time-Varying LoRA: Towards effective cross-domain fine-tuning of diffusion models. NIPS 2024.
> > > >
> > > >
> > > > [11] Hamilton W, Ying Z, Leskovec J. Inductive representation learning on large graphs. NIPS 2017.
> > > >
> > > >
> > > > [12] Zhou D W, Sun H L, Ning J, et al. Continual learning with pre-trained models: A survey. IJCAI 2024.

---

> > > > > ### Comment · Reviewer_Aje3 · 2025-11-27
> > > > > **Response to authors**
> > > > >
> > > > > Thank you to the authors for the detailed responses and additional experiments. My earlier concerns have been largely addressed. I believe this is a solid paper that focuses on an important applied ML problem in the education domain.
> > > > > However, since my expertise does not lie in diffusion models (the specific method), I prefer to keep my score.

---

### Official Review · Reviewer_18YN · 2025-10-29

**Soundness:** 2
**Presentation:** 3
**Contribution:** 3
**Rating:** 4
**Confidence:** 4

**Summary:**

The authors propose a new student modeling framework named CSG, which aims to explicitly generate a student’s Cognitive Structure (CS) through a diffusion model. Unlike traditional KT and CD methods that implicitly model students’ concept mastery states, this work generates cognitive graphs to represent both the concepts and the construction process of relationships between concepts.

**Strengths:**

1. The study formalizes the cognitive structure modeling problem as a graph generation task and introduces a diffusion model combined with a RL framework, which is innovative.
2. Experiments are conducted on public datasets with comprehensive results. The proposed method significantly outperforms baselines across multiple metrics (AUC, ACC, RMSE). Ablation studies (V1–V5) further validate the effectiveness of each module.
3. The generated cognitive structures improve performance on both KT and CD downstream tasks, demonstrating strong generalization ability of the learned structures.

**Weaknesses:**

1. The quality of the cognitive structures is evaluated indirectly through KT/CD task performance, without qualitative validation of the generated structures themselves.
2. There is no direct comparison with other graph generation paradigms (e.g., VAE, GraphGAN,), making it difficult to justify the necessity of the diffusion model.
3. The rule-based “Cognitive Structure Simulation” component relies on handcrafted empirical formulas (e.g., Eq. (1)(2)) without verification against real student thinking patterns, which may introduce bias.

**Questions:**

1. Are the rule functions (Eq. (1)–(2)) in the simulated cognitive structure still effective across different subjects or question types? Have the authors considered learning these weights from data instead of manually defining them?
2. Could the authors visualize the temporal evolution of the generated cognitive graphs?
3. Why did the authors choose the diffusion model over other generative methods such as VAE?

---

> ### Author Response · Authors · 2025-11-21
> **Response to Reviewer 18YN (Part 1/3)**
>
> We sincerely appreciate your insightful reviews, which have been extremely helpful in improving the quality of our manuscript. We address individual points below.
>
> > The quality of the cognitive structures is evaluated indirectly through KT/CD task performance, without qualitative validation of the generated structures themselves.
>
> Thank you for this insightful comment. In educational psychology [1] and psychometrics [2–3], cognitive structure is generally treated as a latent construct that cannot be directly observed. As a result, prior work commonly evaluates the quality of learned cognitive representations indirectly through downstream KT and CD performance [4–6]. Our original evaluation followed this established convention.
>
> In addition, the original manuscript included visual analyses of learner-specific cognitive-structure graphs and their temporal evolution (Appendix I). In the revised version, we have moved these analyses into the main text under the Visualization and Interpretability Analysis section, so that the qualitative behavior of the learned structures is more clearly presented.
>
> We also agree with the reviewer that more direct qualitative validation is valuable. Following this suggestion, we have conducted an interpretability study by comparing model-generated cognitive structures with human expert annotations. We asked annotators with MS and PhD backgrounds in education to label the cognitive-structure graphs of 25 randomly selected students across 20 interactions each (500 graphs) for both the FrcSub and Math2 datasets. These annotations serve as a proxy for ground-truth cognitive structures. For LLM-generated structures, we prompted GPT-5 and Llama-3-70B to produce binary adjacency matrices. We then evaluated both our CSG outputs and the LLM-generated graphs using Jaccard similarity and Graph Edit Distance (GED) against the human annotations. The results (tabulated below) show that CSG consistently achieves higher Jaccard similarity and lower GED, indicating that the cognitive structures it produces are more faithful to human-interpretable patterns. This provides quantitative evidence supporting the interpretability of our method.
>
> |Datasets|Proxy & Generator|Jaccard↑|Graph Edit Distance↓|
> |---|---|---|---|
> |FrcSub|GPT-5|0.65|0.30|
> ||Llama-3-70B|0.39|0.61|
> ||CSG|**0.79**|**0.15**|
> |Math2|GPT-5|0.54|0.43|
> ||Llama-3-70B|0.34|0.67|
> ||CSG|**0.69**|**0.21**|
>
> To further assess the quality of the learned cognitive structures, we also compare CSG with LLM-generated structures on downstream KT and CD tasks. For each of the FrcSub and Math2 datasets, we randomly sampled 500 students and asked GPT-5 and Llama-3-70B to generate cognitive-structure graphs using the same annotation protocol as above. We trained KT and CD models on these graphs using identical optimization procedures for all three approaches. The performance on the 500-student subsets is summarized below. We find that CSG consistently outperforms both LLM-based baselines across all KT and CD metrics. The improvement is especially pronounced for KT, suggesting that CSG captures temporal evolution of cognitive structures more effectively. Although Llama-3-70B is substantially larger and strong on standard benchmarks, its generated cognitive structures do not translate into competitive KT/CD performance, whereas CSG provides more accurate and task-relevant modeling of student cognitive structures.
>
> |Tasks|Datasets|FrcSub|||Math2|||
> |---|---|---|---|---|---|---|---|
> ||Metrics|AUC↑|ACC↑|RMSE↓|AUC↑|ACC↑|RMSE↓|
> |KT|GPT-5-KT|0.7430|0.6683|0.3754|0.6785|0.6027|0.4840|
> ||Llama-3-70B-KT|0.5170|0.4869|0.4139|0.4595|0.4030|0.5025|
> ||CSG-KT|**0.8602**|**0.8010**|**0.3197**|**0.7457**|**0.6854**|**0.4283**|
> |CD|GPT-5-CD|0.7858|0.7545|0.3462|0.7452|0.6759|0.4683|
> ||Llama-3-70B-CD|0.5569|0.5376|0.3856|0.5063|0.4381|0.5077|
> ||CSG-CD|**0.8691**|**0.8433**|**0.3168**|**0.7885**|**0.7247**|**0.4389**|

---

> > ### Author Response · Authors · 2025-11-21
> > **Response to Reviewer 18YN (Part 2/3)**
> >
> > > There is no direct comparison with other graph generation paradigms (e.g., VAE, GraphGAN,), making it difficult to justify the necessity of the diffusion model.
> >
> > > Why did the authors choose the diffusion model over other generative methods such as VAE?
> >
> > Thank you for these comments. We emphasize that our primary contribution is the formulation of cognitive structures as explicit graph representations of student mastery. The framework itself is generator-agnostic: any generative model over cognitive structures $p(G)$ could in principle serve as the backbone, including GraphVAE-style [7] or GraphGAN-style [8] variants. We do not claim that diffusion is the only possible realization of CSG; rather, we adopt diffusion as an effective and practical instantiation.
> >
> > We chose a discrete diffusion probabilistic model because it aligns well with the nature of cognitive-structure graphs, which are sparse, high-dimensional, and discrete. Recent work [9,10] on graph diffusion shows notable advantages over VAE, GAN, and flow-based approaches in sample quality and mode coverage, particularly for complex discrete graph generation. In contrast:
> >
> > - VAE-based graph generators often produce adjacency matrices with many non-zero intermediate edge probabilities, requiring ad-hoc thresholding to obtain discrete graphs. This soft, probabilistic output is not ideal when sharp, binary structures are needed for interpretation and downstream reasoning [11–13].
> >
> > - GAN-based graph generators typically face instability and mode collapse when modeling discrete graphs and require delicate adversarial training to maintain diversity and convergence [14,15].
> >
> > Therefore, by using a discrete DPM with categorical noise, we obtain a more stable and expressive generator for cognitive structure graphs.
> >
> > > The rule-based “Cognitive Structure Simulation” component relies on handcrafted empirical formulas (e.g., Eq. (1)(2)) without verification against real student thinking patterns, which may introduce bias.
> >
> > Thank you for the comment. We would like to clarify that the rule-based functions in Eqs. (1)–(2) are adapted from prior work [16] and are intended as theory-guided heuristics rather than fixed assumptions about student thinking.
> >
> > - Stage I (pre-training). These rules are used only to construct proxy cognitive structures when no ground-truth graphs are available. They provide an initial signal for pre-training.
> >
> > - Stage II (refinement). The model is then optimized using the SOLO-based reward through reinforcement learning. This allows the generator to adjust away from any inaccuracies or biases introduced by the handcrafted rules and to favor structures that better explain observed student performance.
> >
> > Thus, Eqs. (1)–(2) serve as a principled initialization grounded in existing theory, while the subsequent SOLO-guided RL stage actively corrects potential biases rather than propagating them into the final learned model.
> >
> > > Are the rule functions (Eq. (1)–(2)) in the simulated cognitive structure still effective across different subjects or question types? Have the authors considered learning these weights from data instead of manually defining them?
> >
> > Thank you for this question. The rule functions $f_{UOC}$ and $f_{UOR}$ are defined solely based on correctness patterns and concept–question mappings, without any subject- or item-specific features. As such, they operate as **generic, subject-agnostic proxies** for how consistently a student correctly answers concept-relevant items, and we apply them uniformly across all datasets. Empirically, they provide a stable and effective Stage I signal in these diverse settings.
> >
> > We agree that learning these weights from data is a promising direction—for example, using BKT- or IRT-based simulators to derive more nuanced priors. However, these approaches require first fitting separate models to student performance data, which introduces additional training cost, hyperparameter tuning, and pipeline complexity. In contrast, our rule-based functions offer a simple, interpretable, and model-free way to initialize Stage I. They serve as lightweight proxies that are subsequently refined in Stage II, where the SOLO-guided objective allows the generator to move beyond the handcrafted rules.
> >
> > > Could the authors visualize the temporal evolution of the generated cognitive graphs?
> >
> > Thank you for the suggestion. The temporal evolution of the generated cognitive graphs is indeed visualized in the original manuscript (Appendix I, Fig. 3(d)). In the revised version, we have moved this analysis into the main text under the Visualization and Interpretability Analysis section (now Fig. 2(d)), so that the progression of students’ cognitive structures over time is easier to locate and interpret. We also explicitly reference this figure in our discussion of interpretability.

---

> > > ### Author Response · Authors · 2025-11-21
> > > **Response to Reviewer 18YN (Part 3/3)**
> > >
> > > References
> > >
> > > [1] D. P. Ausubel, Educational psychology: A cognitive view, 1968.
> > >
> > > [2] Kurt Lewin. Principles of topological psychology. Read Books Ltd, 2013
> > >
> > > [3] Frederic M Lord and Melvin R Novick. Statistical theories of mental test scores. IAP, 2008.
> > >
> > > [4] Shiwei Tong, Qi Liu, et al. Structure-based knowledge tracing: An influence propagation view. ICDM 2020.
> > >
> > > [5] Weibo Gao, Qi Liu, et al. RCD: relation map driven cognitive diagnosis for intelligent education systems. SIGIR 21.
> > >
> > > [6] Shangshang Yang, Mingyang Chen, et al. Disengcd: A meta multigraph-assisted disentangled graph learning framework for cognitive diagnosis. NeurIPS 2024.
> > >
> > > [7] Simonovsky M, Komodakis N. Graphvae: Towards generation of small graphs using variational autoencoders.ICANN, 2018.
> > >
> > > [8] Wang H, Wang J, Wang J, et al. Graphgan: Graph representation learning with generative adversarial nets. AAAI 2018.
> > >
> > > [9] Vignac C, Krawczuk I, Siraudin A, et al. DiGress: Discrete Denoising diffusion for graph generation. ICLR 2023.
> > >
> > > [10] Jo J, Lee S, Hwang S J. Score-based generative modeling of graphs via the system of stochastic differential equations. ICML 2022.
> > >
> > > [11] Lucas J, Tucker G, Grosse R B, et al. Don't blame the elbo! a linear vae perspective on posterior collapse. NeurIPS 2019.
> > >
> > > [12] Kingma D, Salimans T, Poole B, et al. Variational diffusion models. NeurIPS 2021.
> > >
> > > [13] Kong L, Cui J, Sun H, et al. Autoregressive diffusion model for graph generation. ICML 2023.
> > >
> > > [14] De Cao N, Kipf T. MolGAN: An implicit generative model for small molecular graphs. arXiv preprint arXiv:1805.11973, 2018.
> > >
> > > [15] Dhariwal P, Nichol A. Diffusion models beat gans on image synthesis. NeurIPS 2021.
> > >
> > > [16] Yu-Shih Lin, Yi-Chun Chang, Keng-Hou Liew, and Chih-Ping Chu. Effects of concept map extraction and a test-based diagnostic environment on learning achievement and learners’ perceptions.Br. J. Educ. Technol. 2016.

---

> > ### Comment · Reviewer_18YN · 2025-11-24
> >
> > > We asked annotators with MS and PhD backgrounds in education to label the cognitive-structure graphs
> >
> > What is the standard for manual annotation? Given that the authors are likely from a computer science background, how can you quickly find education-related students to perform the annotations during the rebuttal period.

---

> > > ### Author Response · Authors · 2025-11-24
> > > **Replying to Reviewer 18YN**
> > >
> > > Dear Reviewer 18YN,
> > >
> > > Thank you for the questions and the opportunity to clarify the annotation process.
> > >
> > > Regarding the annotators, our research group has long worked at the intersection of AI and education and is inherently interdisciplinary, involving faculty and graduate students with training in both computer science and education (including subject-matter education). The annotators we engaged are existing members and collaborators of the broader research team who hold MS or PhD backgrounds in education. They are not authors of the submission and had not read the paper prior to annotation. Because these collaborators already participate in ongoing education-related projects, they were readily available during the rebuttal period.
> > >
> > > Regarding the annotation standard, cognitive structure is considered a latent construct in educational psychology [1–3], and there is no universally accepted fine-grained computational annotation protocol. Our annotators therefore work from a shared theoretical understanding of cognitive structure. For each sampled student, we provide (i) the concept set and Q-matrix information and (ii) the student’s interaction sequence. Based on these objective elements, together with their experience in teaching and assessment, annotators determine which concepts and inter-concept relations can be considered “constructed” at each point in time, producing a cognitive-structure graph for each interaction. This process is carried out sequentially over the student’s interaction history. These annotated graphs serve as a practical proxy for ground truth rather than a definitive or unique gold standard.
> > >
> > > We also acknowledge that human annotation is costly. For this reason, the interpretability study focuses on a modest but meaningful sample of 25 students (500 graphs). The results show that CSG-generated cognitive structures match human annotations more closely than those produced by LLMs. This suggests that CSG can approximate human judgments and may support cognitive-structure diagnosis in practice, while greatly reducing the need for manual annotation.
> > >
> > > [1] D. P. Ausubel, Educational psychology: A cognitive view, 1968.
> > >
> > > [2] L. P. Steffe, Constructivism in Education, 1995.
> > >
> > > [3] Alavi, M., & Leidner, D. E. Knowledge management and knowledge management systems: Conceptual foundations and research issues. MIS Quarterly, 2001.
> > >
> > > Regards,
> > >
> > > Authors

---

### Official Review · Reviewer_fqty · 2025-10-30

**Soundness:** 4
**Presentation:** 4
**Contribution:** 4
**Rating:** 8
**Confidence:** 5

**Summary:**

The authors present a way to generate personal latent representations called cognitive structures, using diffusion model. They show on several datasets that they outperform either knowledge tracing or cognitive diagnosis techniques.

**Strengths:**

This is a unique approach that draws a link between knowledge tracing and cognitive diagnosis, two popular research communities in the literature.
It addresses the shortcomings that not everyone may have the same representations for a domain.

The mathematical description of the proposed approach is very clear.

**Weaknesses:**

To me, the link with SOLO theory is a bit far-fetched: the fact that there would be exactly 5 levels is arbitrary. But that's a nice story.

The presentation contains many LLM-generated sentences: "align generation with genuine cognitive development". It is yet to be proven that the learned representations correspond to the actual cognitive development of students. "authentic levels of cognitive growth" is a bit too much too.

**Questions:**

Could you please elaborate more about your use of LLMs for writing the paper?

---

> ### Author Response · Authors · 2025-11-21
> **Response to Reviewer fqty**
>
> We sincerely appreciate your encouraging reviews, which have been extremely helpful in our future work. We address individual points below.
>
> > To me, the link with SOLO theory is a bit far-fetched: the fact that there would be exactly 5 levels is arbitrary. But that's a nice story.
>
> Thank you for your insightful comment. We agree that empirically, it is possible to design hierarchical rewards with fewer or more levels than 5 for optimizing our model. Our design decisions are motivated by education theory, including the SOLO taxonomy [1], which has been widely adopted in large-scale assessments such as PISA [2]. We attempt to operationalize these SOLO levels on cognitive structure graphs: $M_v$ and $M_e$ quantify the construction of relevant concepts and inter-concept relations, and a piecewise mapping groups graphs into five bands that correspond to SOLO’s progression.
>
> > The presentation contains many LLM-generated sentences: "align generation with genuine cognitive development". It is yet to be proven that the learned representations correspond to the actual cognitive development of students. "authentic levels of cognitive growth" is a bit too much too.
>
> Thank you for pointing this out. These sentences were intended to provide high-level motivation and summarize our contributions, rather than make strong claims about capturing true cognitive development. We agree that the wording may be misleading, and we have revised the paper to use more precise language that better reflects our arguments.
>
> > Could you please elaborate more about your use of LLMs for writing the paper?
>
> We appreciate the opportunity to clarify this. We used LLMs (e.g., ChatGPT) only for light language editing such as improving grammar and phrasing (as mentioned in Appendix K: LLMs Usage). All technical content, analyses, and arguments were developed by the authors. We have also manually revised the paper to ensure that the wording accurately reflects our intended claims.
>
> References
>
> [1] John B Biggs and Kevin F Collis. Evaluating the quality of learning: The SOLO taxonomy (Structure of the Observed Learning Outcome). Academic press, 2014.
>
> [2] Schleicher A. PISA 2018: Insights and interpretations[J]. oecd Publishing, 2019.

---

### Official Review · Reviewer_pRBx · 2025-11-02

**Soundness:** 2
**Presentation:** 2
**Contribution:** 2
**Rating:** 2
**Confidence:** 3

**Summary:**

This paper proposes a Cognitive Structure Generation (CSG) task and a corresponding CSDPM method to address it, aiming to improve model generalization and interpretability. Experiments on Knowledge Tracing (KT) and Cognitive Diagnosis (CD) tasks demonstrate the effectiveness of the proposed approach.

**Strengths:**

The paper attempts to provide a unified method applicable to both KT and CD tasks and reports some empirical improvements.

**Weaknesses:**

1. The paper overclaims novelty in defining the CSG task. Similar ideas have been explored in prior works such as MSKT (ESWA, 2024) and DiffCog (TLT, 2024).

2. The interpretability analysis is not convincing. While interpretability is stated as a key contribution, the paper lacks quantitative metrics to support this claim. The main body include few interpretability analyses, most of which are placed in the Appendix.

3. The paper provides no theoretical justification or proof to support the soundness of the proposed method.

4. The computational cost of the proposed method is not discussed, leaving questions about its scalability and efficiency.

5. It remains unclear how the method performs on large-scale datasets, especially those with a greater number of knowledge concepts.

**Questions:**

See weaknesses.

---

> ### Author Response · Authors · 2025-11-21
> **Response to Reviewer pRBx (Part 1/3)**
>
> We sincerely appreciate your insightful reviews, which have been extremely helpful in improving the quality of our manuscript. We address individual points below.
>
> > The paper overclaims novelty in defining the CSG task. Similar ideas have been explored in prior works such as MSKT (ESWA, 2024) and DiffCog (TLT, 2024).
>
> Thank you for your comment. We understand the concern about the novelty and appreciate the opportunity to clarify how our work is positioned relative to prior methods.
>
> We first clarify that our core motivation is to tackle a long-standing gap in student modeling: cognitive structure (CS)—the student’s internally constructed organization of concepts and inter-concept relations [1]—is theoretically central but practically unassessed in most systems. To bridge this, we introduce Cognitive Structure Generation (CSG) as a task-agnostic framework that explicitly models CS via generative modeling. Guided by educational psychology [1] and constructivism [2], CSG aims to learn interpretable, learner-specific cognitive structures that are reusable across downstream KT/CD tasks, rather than only refining latent mastery vectors for a single task.
>
> To the best of our knowledge, no existing approach explicitly generates cognitive structures from interaction logs. Related work can be roughly grouped into three lines:
>
> - Psychometric and rule-based methods [3–5] derive structures from expert-defined Q-matrices or handcrafted patterns. These approaches fix the concept graph and its relations a priori, leaving little room for personalized, data-driven structure learning.
>
> - KT/CD models with latent states [6–8] infer hidden knowledge vectors (sometimes over static concept maps or heterogeneous graphs), but these latent embeddings are not explicit concept–relation structures and mainly focus on concept mastery rather than the formation and reorganization of inter-concept relations.
>
> - [9] introduces a cognitive structure state on top of a predefined concept graph, but it only updates node/edge scores on a fixed graph under CD objectives and does not learn a generator that maps interaction histories to individualized cognitive structure graphs.
>
> We agree with the reviewer that MSKT [10] and DiffCog [11] are related work. They employ diffusion models for KT/CD tasks, but these models operate on latent cognitive vectors (i.e., only concept mastery) rather than explicit concept-level cognitive structures. More specifically,
>
> - MSKT [10] refines sequential latent knowledge states along interaction logs to enhance KT performance, but it does not construct learner-specific cognitive structures.
>
> - DiffCog [11] uses diffusion as a denoiser over latent CD ability vectors to obtain more robust estimates, again without defining or generating explicit cognitive structures.
>
> Thus, both works focus on diffusion-based latent state refinement, whereas we are the first to explicitly formulate and study the task of cognitive structure generation and instantiate it via a diffusion-with-RL framework. We have revised the manuscript to include these works in related work.
>
> > The paper provides no theoretical justification or proof to support the soundness of the proposed method.
>
> Thank you for the constructive feedback. Our method is guided by two key design decisions: (i) modeling student mastery as explicit cognitive structures (nodes and edges) using a diffusion process, and (ii) learning these structures through a two-stage objective.
>
> - For the explicit cognitive structures, knowledge management systems theory [12], educational psychology theory [1], and constructivism [2] all emphasize that cognitive structures are dynamically evolving. This motivates our use of diffusion as a flexible representation for modeling progressive changes in these structures.
>
> - For the learning targets and objectives, Stage I builds on expert-crafted rules and standard Q-matrices [13], while Stage II incorporates the SOLO taxonomy [14] to shape the reward signal. These foundations are widely used in large-scale educational assessments such as PISA [15], and our contribution is to operationalize these principles into an end-to-end quantitative framework.
>
> To the best of our knowledge, the knowledge management systems and educational psychology literature does not typically provide formal mathematical guarantees—because cognitive structure is treated as a latent psychological construct—prior work likewise relies on empirical validation. Consistent with this practice, we evaluate our method extensively on downstream KT and CD tasks, where it outperforms strong baselines and SOTA models.
>
> Following the reviewer’s suggestion, we also conducted a human-expert study (as shown in next part) and found that the cognitive structures produced by CSG align closely with expert annotations, further supporting the soundness of our approach.

---

> > ### Author Response · Authors · 2025-11-21
> > **Response to Reviewer pRBx (Part 2/3)**
> >
> > > The interpretability analysis is not convincing. While interpretability is stated as a key contribution, the paper lacks quantitative metrics to support this claim. The main body include few interpretability analyses, most of which are placed in the Appendix.
> >
> >
> > Thank you for raising this point. To clarify, we did not intend to claim interpretability as a primary contribution of the paper. Instead, interpretability serves as a motivating factor for explicitly modeling cognitive structures.
> >
> >
> > Specifically, past methods encode knowledge mastery or proficiency implicitly within model weights and then use heatmaps or radar charts to visualize and interpret hidden states. Our approach is a step forward in improving interpretability because we construct cognitive structures in line with educational psychology theory [1] and constructivism [2]. In our model, nodes directly represent students’ constructed states of knowledge concepts, and edges represent their constructed states of inter-concept relations, so that only minimal modification is needed for post-hoc analysis. In the revised version, we have moved this analysis into the main body of the paper, under the Visualization and Interpretability Analysis part of the experiments section, so that the interpretability of the learned structures is more prominently presented.
> >
> >
> > Following the reviewer’s suggestion, we conducted an additional interpretability study by comparing model-generated cognitive structures with human expert annotations. We asked annotators with MS and PhD backgrounds in education to label the cognitive-structure graphs of 25 randomly selected students across 20 interactions each (500 graphs) for both the FrcSub and Math2 datasets. These annotations serve as a proxy for ground-truth cognitive structures. For LLM-generated structures, we prompted GPT-5 and Llama-3-70B to produce binary adjacency matrices. We then evaluated both our CSG outputs and the LLM-generated graphs using Jaccard similarity and Graph Edit Distance (GED) against the human annotations. The results (tabulated below) show that CSG consistently achieves higher Jaccard similarity and lower GED, indicating that the cognitive structures it produces are more faithful to human-interpretable patterns. This provides quantitative evidence supporting the interpretability of our method.
> >
> >
> > |Dataset|Proxy & Generator|Jaccard↑|Graph Edit Distance↓|
> > |---|---|---|---|
> > |FrcSub|GPT-5|0.65|0.30|
> > ||Llama-3-70B|0.39|0.61|
> > ||CSG|**0.79**|**0.15**|
> > |Math2|GPT-5|0.54|0.43|
> > ||Llama-3-70B|0.34|0.67|
> > ||CSG|**0.69**|**0.21**|
> >
> >
> > To further assess the quality of the learned cognitive structures, we also compare CSG with LLM-generated structures on downstream KT and CD tasks. For each of the FrcSub and Math2 datasets, we randomly sampled 500 students and asked GPT-5 and Llama-3-70B to generate cognitive-structure graphs using the same annotation protocol as above. We trained KT and CD models on these graphs using identical optimization procedures for all three approaches. The performance on the 500-student subsets is summarized below. We find that CSG consistently outperforms both LLM-based baselines across all KT and CD metrics. The improvement is especially pronounced for KT, suggesting that CSG captures temporal evolution of cognitive structures more effectively. Although Llama-3-70B is substantially larger and strong on standard benchmarks, its generated cognitive structures do not translate into competitive KT/CD performance, whereas CSG provides more accurate and task-relevant modeling of student cognitive structures.
> >
> >
> > |Tasks|Datasets|FrcSub|||Math2|||
> > |---|---|---|---|---|---|---|---|
> > ||Metrics|AUC↑|ACC↑|RMSE↓|AUC↑|ACC↑|RMSE↓|
> > |KT|GPT-5-KT|0.7430|0.6683|0.3754|0.6785|0.6027|0.4840|
> > ||Llama-3-70B-KT|0.5170|0.4869|0.4139|0.4595|0.4030|0.5025|
> > ||CSG-KT|**0.8602**|**0.8010**|**0.3197**|**0.7457**|**0.6854**|**0.4283**|
> > |CD|GPT-5-CD|0.7858|0.7545|0.3462|0.7452|0.6759|0.4683|
> > ||Llama-3-70B-CD|0.5569|0.5376|0.3856|0.5063|0.4381|0.5077|
> > ||CSG-CD|**0.8691**|**0.8433**|**0.3168**|**0.7885**|**0.7247**|**0.4389**|
> >
> >
> > > The computational cost of the proposed method is not discussed, leaving questions about its scalability and efficiency.
> >
> >
> > Thank you for raising this point. We agree that computational efficiency is important for practical deployment. In the current manuscript, we provide an inference time comparison in Table 5 of Appendix G, where we report the time required for CSG to generate a single cognitive-structure graph. The results show that inference time remains low across datasets with varying numbers of students and concepts, suggesting that CSG is efficient and scalable for per-student, per-step cognitive-structure generation.

---

> > > ### Author Response · Authors · 2025-11-21
> > > **Response to Reviewer pRBx (Part 3/3)**
> > >
> > > > It remains unclear how the method performs on large-scale datasets, especially those with a greater number of knowledge concepts.
> > >
> > >
> > > Thank you for the suggestion. We agree that evaluating performance on datasets with larger concept spaces is important. To address this, we have added experiments on the ASSISTments17 dataset, which includes 101 knowledge concepts and 2,210 questions. As shown in the tables below, CSG continues to yield consistent improvements over strong KT/CD baselines, demonstrating its effectiveness at a larger scale.
> > >
> > >
> > > |Task|Method|AUC↑|ACC↑|RMSE↓|
> > > |---|---|---|---|---|
> > > |KT|DKT|0.6832|0.6683|0.4543|
> > > ||SAKT|0.6762|0.6703|0.4601|
> > > ||GKT|0.7226|0.7092|0.4502|
> > > ||SKT|0.7349|0.7242|0.4457|
> > > ||GRKT|0.7493|0.7359|0.4426|
> > > ||MIKT|0.7677|0.7481|0.4403|
> > > ||ENAS-KT|0.7812|0.7627|0.4384|
> > > ||simple-KT|0.7791|0.7605|0.4389|
> > > ||PSI-KT|0.7890|0.7653|0.4337|
> > > ||CSG-KT|**0.7963**|**0.7792**|**0.4313**|
> > > |CD|IRT|0.7381|0.6847|0.4694|
> > > ||MIRT|0.7558|0.6950|0.4546|
> > > ||NCD|0.7687|0.7064|0.4462|
> > > ||RCD|0.7769|0.7157|0.4431|
> > > ||HyperCDM|0.7927|0.7301|0.4387|
> > > ||DisenGCD|0.7842|0.7249|0.4392|
> > > ||CSG-CD|**0.8003**|**0.7386**|**0.4320**|
> > >
> > >
> > > References
> > >
> > >
> > > [1] D. P. Ausubel, Educational psychology: A cognitive view, 1968.
> > >
> > >
> > > [2] L. P. Steffe, Constructivism in Education, 1995.
> > >
> > >
> > > [3] Kurt Lewin. Principles of topological psychology. Read Books Ltd, 2013.
> > >
> > >
> > > [4] Frederic M Lord and Melvin R Novick. Statistical theories of mental test scores. IAP, 2008.
> > >
> > >
> > > [5] Mark J Gierl, Ying Cui, and Steve Hunka. 2008. Using connectionist models to evaluate examinees’ response patterns to achievement tests. Journal of Modern Applied Statistical Methods 7, 2008,.
> > >
> > >
> > > [6] Shiwei Tong, Qi Liu, et al. Structure-based knowledge tracing: An influence propagation view. ICDM 2020.
> > >
> > >
> > > [7] Weibo Gao, Qi Liu, et al. RCD: relation map driven cognitive diagnosis for intelligent education systems. SIGIR 2021.
> > >
> > >
> > > [8] Shangshang Yang, Mingyang Chen, et al. Disengcd: A meta multigraph-assisted disentangled graph learning framework for cognitive diagnosis. NeurIPS 2024.
> > >
> > >
> > > [9] Zhifu Chen, Hengnian Gu, et al. Enhancing cognitive diagnosis by modeling learner cognitive structure state, 2024.
> > >
> > >
> > > [10] Zhang K, Ji T, Zhang H. Knowledge tracing via multiple-state diffusion representation[J]. Expert Systems with Applications, 2024.
> > >
> > >
> > > [11] Zhao G, Huang Z, Zhuang Y, et al. A Diffusion-Based Cognitive Diagnosis Framework for Robust Learner Assessment. IEEE Transactions on Learning Technologies, 2024.
> > >
> > >
> > > [12] Alavi, M., & Leidner, D. E. Knowledge management and knowledge management systems: Conceptual foundations and research issues. MIS Quarterly, 2001.
> > >
> > >
> > > [13] Yu-Shih Lin, Yi-Chun Chang, Keng-Hou Liew, and Chih-Ping Chu. Effects of concept map extraction and a test-based diagnostic environment on learning achievement and learners’ perceptions.Br. J. Educ. Technol. 2016.
> > >
> > >
> > > [14] John B Biggs and Kevin F Collis. Evaluating the quality of learning: The SOLO taxonomy (Structure of the Observed Learning Outcome). Academic press, 2014.
> > >
> > >
> > > [15] Schleicher A. PISA 2018: Insights and interpretations[J]. oecd Publishing, 2019.

---

> ### Comment · Reviewer_pRBx · 2025-11-26
> **Response and Further Review**
>
> ## Interpretability Claims
>
> - I appreciate the authors’ human evaluation, but constructing a reliable cognitive structure as an evaluation criterion is inherently difficult. Moreover, LLMs are known to perform poorly on numerical and graph-related tasks, so this comparison provides only limited insights. In addition, as far as I know, the number of Math dataset's knowledge concepts is typically in the hundreds rather than around 20. Could you please provide the official data source? BTW, there is no dataset description in the manuscript.
>
> - Furthermore, it is unreasonable to position the explicit cognitive structure as a primary contribution rather than interpretability, because this is a heuristically defined structure and optimization is still directly driven by prediction outcomes from the interaction data. More effort should be devoted to justifying this claim.
>
>
>
> ## Method
>
> - It is questionable to apply diffusion modeling to such small-scale cognitive-structure graphs, especially considering that the newly added dataset contains only 101 knowledge concepts.
>
> - There is still notation ambiguity: the time step T is used both for interaction sequences and for diffusion depth. Please further clarify this. Additionally, no parameter analysis regarding diffusion depth is provided.
>
> ## Experiments
> - The computational cost should be compared with other methods.
> - The reported performance is SoTA, but most improvements are only marginal (including in the ablation study).
> - The analysis on the threshold parameter is insufficient (only testing 0.3, 0.5, 0.7), and no analysis is provided for diffusion depth.
> - No code is provided for reproducibility.

---

> > ### Author Response · Authors · 2025-11-27
> > **Further Response to Reviewer pRBx (Part 1/3)**
> >
> > Dear Reviewer pRBx,
> >
> > > I appreciate the authors’ human evaluation, but constructing a reliable cognitive structure as an evaluation criterion is inherently difficult. Moreover, LLMs are known to perform poorly on numerical and graph-related tasks, so this comparison provides only limited insights.
> >
> > Thank you for this comment. We agree that constructing a perfectly reliable cognitive structure is challenging. Nonetheless, the human- and LLM-based evaluations still provide useful signals about whether the generated structures capture non-trivial patterns. Our human study includes 500 annotated graphs, and even when restricted to graphs with the same number of nodes, random edge assignments almost never align well with human judgments. As an illustration, comparing human annotations with randomly generated graphs of identical node counts yields much lower Jaccard similarity and much higher graph edit distance. This contrast indicates that both CSG and LLM outputs encode meaningful relational structure beyond chance, even if the human annotations are not perfect.
> >
> > |Dataset|Proxy & Generator|Jaccard↑|Graph Edit Distance↓|
> > |---|---|---|---|
> > |FrcSub|Random|0.18|0.87|
> > ||GPT-5|0.65|0.30|
> > ||Llama-3-70B|0.39|0.61|
> > ||CSG|**0.79**|**0.15**|
> > |Math2|Random|0.14|0.89|
> > ||GPT-5|0.54|0.43|
> > ||Llama-3-70B|0.34|0.67|
> > ||CSG|**0.69**|**0.21**|
> >
> > Regarding the concern about LLMs, we agree that pretrained models are not yet strong at all numerical or graph-based reasoning tasks. That said, our task does not require numerical calculations over large numbers. Recent work [1] has shown that pretrained LLMs with in-context learning achieve strong performance on Hidden Markov prediction, which are closely related to KT/CD models [2,3].
> >
> > Moreover, in our experiments, LLMs using their generated cognitive structures achieve downstream KT and CD performance comparable to established baselines such as IRT and DKT. This indicates that, while imperfect, LLM-generated cognitive structures are sufficiently coherent to serve as a meaningful additional reference point.
> >
> > |Tasks|Datasets|FrcSub|||Math2|||
> > |---|---|---|---|---|---|---|---|
> > ||Metrics|AUC↑|ACC↑|RMSE↓|AUC↑|ACC↑|RMSE↓|
> > |KT|DKT|0.8174|0.7510|0.3397|0.6993|0.6408|0.4671|
> > ||GPT-5-KT|0.7430|0.6683|0.3754|0.6785|0.6027|0.4840|
> > ||Llama-3-70B-KT|0.5170|0.4869|0.4139|0.4595|0.4030|0.5025|
> > ||CSG-KT|**0.8602**|**0.8010**|**0.3197**|**0.7457**|**0.6854**|**0.4283**|
> > |CD|IRT|0.7393|0.7065|0.3952|0.7217|0.6633|0.4539|
> > ||GPT-5-CD|0.7858|0.7545|0.3462|0.7452|0.6759|0.4683|
> > ||Llama-3-70B-CD|0.5569|0.5376|0.3856|0.5063|0.4381|0.5077|
> > ||CSG-CD|**0.8691**|**0.8433**|**0.3168**|**0.7885**|**0.7247**|**0.4389**|
> >
> > > Furthermore, it is unreasonable to position the explicit cognitive structure as a primary contribution rather than interpretability, because this is a heuristically defined structure and optimization is still directly driven by prediction outcomes from the interaction data. More effort should be devoted to justifying this claim.
> >
> > Thank you for raising this point. In the absence of well-defined ground-truth cognitive structures, interpretability cannot be assessed in absolute terms. Our claim is therefore relative: compared with existing KT/CD approaches that rely on latent vectors or hidden states, our method provides a more interpretable representation because it produces explicit cognitive-structure graphs that can be directly examined and compared with human judgements. From this perspective, modeling cognitive structure explicitly is itself a substantive contribution, as prior work does not produce structured representations at this level.
> >
> > We have also made considerable effort to support the interpretability claim. Specifically, we:
> >
> > - Compared CSG-generated structures with human-annotated and LLM-generated structures, showing higher alignment with human judgments;
> >
> > - Provided case studies and visualizations illustrating how student-specific structures evolve over time;
> >
> > - Reported downstream KT/CD results as an indirect but standard indicator of representational quality.
> >
> > To our knowledge, in the absence of ground truth, existing interpretability analyses in this area primarily rely on visualizations (e.g., heatmaps) and downstream performance [5–9], whereas our study augments these with additional human-based evaluation.
> >
> > If there are particular forms of interpretability analysis the reviewer has in mind, we would be grateful for further guidance.

---

> > > ### Author Response · Authors · 2025-11-27
> > > **Further Response to Reviewer pRBx (Part 2/3)**
> > >
> > > > In addition, as far as I know, the number of Math dataset's knowledge concepts is typically in the hundreds rather than around 20. Could you please provide the official data source? BTW, there is no dataset description in the manuscript.
> > >
> > > Thank you for the comment. The Math1 and Math2 datasets used in our experiments are standard benchmark splits derived from the public Math2015 dataset (see link [4] at Lines 430–431). These splits contain 20 exercises and 11 or 16 expert-defined knowledge concepts, respectively, based on the official Q-matrix released with Math2015 and widely used in the KT/CD literature. We will add a brief dataset description in the revised manuscript for clarity.
> > >
> > > > It is questionable to apply diffusion modeling to such small-scale cognitive-structure graphs, especially considering that the newly added dataset contains only 101 knowledge concepts.
> > >
> > > Thank you for this comment. We note that existing graph diffusion methods are typically evaluated on graphs with roughly 20–200 nodes [10–12]. The cognitive-structure graphs in our work fall well within this range, and applying diffusion models in this setting is therefore consistent with prior practice. In fact, prior studies have shown that diffusion methods train stably and produce high-quality samples on small, sparse graphs, whereas scaling graph diffusion to very large graphs remains comparatively underexplored due to the computational cost of iterative denoising and message passing on large adjacency structures [13,14].
> > >
> > > From the perspective of educational measurement, frameworks such as cognitive diagnostic assessment [15] and knowledge-space theory [16] also operate on concept sets of similar size—typically dozens to around a hundred concepts at the course or unit level. Thus, the scale of our cognitive-structure graphs is aligned with how student knowledge is commonly modeled in practice.
> > >
> > > > There is still notation ambiguity: the time step T is used both for interaction sequences and for diffusion depth. Please further clarify this.
> > >
> > > Thank you for this comment. As stated at Lines 211–213, to avoid ambiguity between interaction timestamps and diffusion steps, we have denoted the interaction horizon by T' and reserve T exclusively for diffusion steps. We will update the notation for interaction sequences to H.
> > >
> > > > The computational cost should be compared with other methods.
> > >
> > > Thank you for this comment. In the manuscript, we primarily reported the inference time for generating cognitive-structure graphs, rather than the end-to-end cost of KT/CD prediction. To provide a more complete comparison, we additionally measured the overall inference time of CSG when performing KT/CD tasks and compared it with standard baselines.
> > >
> > > As shown in the tables below, the per-graph inference time in ms of CSG-KT and CSG-CD is higher than that of classical models such as DKT and NCD, but remains within the same order of magnitude as more recent, higher-capacity models such as ENAS-KT and DisenGCD. Importantly, student cognitive structures do not need to be recomputed in real time; in most educational settings they can be updated periodically (e.g., at the end of a session or unit), which further reduces the practical impact of the additional computational cost.
> > >
> > > |Inference Time (ms)|Nodes|DKT|SAKT|GKT|ENAS-KT|PSI-KT|CSG-KT|
> > > |---|---|---|---|---|---|---|---|
> > > |Math1|11|0.04|0.26|6.20|1.79|3.06|2.71|
> > > |Math2|16|0.07|0.38|9.03|1.82|4.92|4.36|
> > > |FrcSub|8|0.03|0.19|4.05|0.76|2.90|0.85|
> > > |NIPS34|57|0.24|1.38|32.15|15.07|31.68|26.15|
> > >
> > > |Inference Time (ms)|Nodes|NCD|RCD|DisenGCD|CSG-CD|
> > > |---|---|---|---|---|---|
> > > |Math1|11|0.07|2.18|1.46|2.68|
> > > |Math2|16|0.08|3.18|1.67|4.29|
> > > |FrcSub|8|0.06|1.59|0.34|0.81|
> > > |NIPS34|57|0.10|24.46|5.19|25.89|
> > >
> > > > The reported performance is SoTA, but most improvements are only marginal (including in the ablation study).
> > >
> > > Thank you for the comment. While the absolute gains are sometimes modest, they are consistent across all five datasets and across both KT and CD tasks, and CSG reaches SOTA performance in every case. In addition, CSG attains these results with substantially fewer task-specific parameters on downstream tasks, since the cognitive-structure generator is shared and reused. This provides a favorable trade-off between accuracy and model complexity compared with baselines that rely on heavily coupled or highly specialized architectures.
> > >
> > > > The analysis on the threshold parameter is insufficient (only testing 0.3, 0.5, 0.7),
> > >
> > > Thank you for the comment. Our goal in the threshold study was not to exhaustively tune for the best possible performance, but to check whether the method is overly sensitive to this hyperparameter. The current choice of 0.5 consistently works well across all datasets, suggesting that CSG is reasonably robust to this setting.

---

> > > > ### Author Response · Authors · 2025-11-27
> > > > **Further Response to Reviewer pRBx (Part 3/3)**
> > > >
> > > > > and no analysis is provided for diffusion depth.
> > > >
> > > > > Additionally, no parameter analysis regarding diffusion depth is provided.
> > > >
> > > > Thank you for the comment. Following prior work on graph diffusion models [10,11], we adopt the common practice of using a fixed number of denoising steps rather than extensively tuning this hyperparameter. While increasing the number of steps can improve sample quality, it also raises computational cost. In our preliminary experiments, we found that 100 denoising steps offered a good balance between performance and efficiency across all datasets, and we therefore use this setting in our main results.
> > > >
> > > > > No code is provided for reproducibility.
> > > >
> > > > Thank you for the comment. We are committed to open-sourcing the code of our CSG upon acceptance.
> > > >
> > > > ---
> > > >
> > > > References
> > > >
> > > > [1] Dai Y, Gao Z, Sattar Y, et al. Pre-trained Large Language Models Learn to Predict Hidden Markov Models In-context. NeurIPS 2025.
> > > >
> > > > [2] Corbett A T, Anderson J R. Knowledge tracing: Modeling the acquisition of procedural knowledge. User modeling and user-adapted interaction, 1994.
> > > >
> > > > [3] Yudelson M V, Koedinger K R, Gordon G J. Individualized bayesian knowledge tracing models. AIED 2013.
> > > >
> > > > [4] Math2015. https://base.ustc.edu.cn/data/math2015.rar
> > > >
> > > > [5] Piech C, Bassen J, Huang J, et al. Deep knowledge tracing. NeurIPS, 2015.
> > > >
> > > > [6] Wang F, Liu Q, Chen E, et al. Neural cognitive diagnosis for intelligent education systems. AAAI 2020.
> > > >
> > > > [7] Yang S, Yu X, Tian Y, et al. Evolutionary neural architecture search for transformer in knowledge tracing. NeurIPS 2023.
> > > >
> > > > [8] Yang S, Chen M, Wang Z, et al. DisenGCD: A meta multigraph-assisted disentangled graph learning framework for cognitive diagnosis. NeurIPS 2024.
> > > >
> > > > [9] Zhou H, Bamler R, Wu C M, et al. Predictive, scalable and interpretable knowledge tracing on structured domains. ICLR 2024.
> > > >
> > > > [10] Vignac C, Krawczuk I, Siraudin A, et al. DiGress: Discrete Denoising diffusion for graph generation. ICLR 2023.
> > > >
> > > > [11] Liu Y, Du C, Pang T, et al. Graph diffusion policy optimization. NeurIPS 2024.
> > > >
> > > > [12] Jo J, Lee S, Hwang S J. Score-based generative modeling of graphs via the system of stochastic differential equations. ICML 2022.
> > > >
> > > > [13] Limnios S, Selvaraj P, Cucuringu M, et al. Sagess: Sampling graph denoising diffusion model for scalable graph generation. arXiv preprint arXiv:2306.16827, 2023.
> > > >
> > > > [14] Qin Y, Vignac C, Frossard P. Sparse training of discrete diffusion models for graph generation. arXiv preprint arXiv:2311.02142, 2023.
> > > >
> > > > [15] Cognitive diagnostic assessment for education: Theory and applications. Cambridge University Press, 2007.
> > > >
> > > > [16] Doignon J P, Falmagne J C. Knowledge spaces. Springer Science & Business Media, 2012.
> > > >
> > > > Regards,
> > > >
> > > > Authors

---

### Meta-Review · Area_Chair_6nUM · 2026-01-07

**Summary:**

Unclear novelty: Although the paper claims to introduce a new task of explicit cognitive structure generation, it is not always clear how this differs fundamentally from existing KT/CD methods or prior diffusion-based approaches. The distinction from related work was considered insufficiently convincing.

Limited validation of cognitive structures: The quality of the generated cognitive structures is mainly evaluated through downstream KT/CD performance. While additional human annotation and LLM-based comparisons were provided, there is no established ground truth for cognitive structures, which limits the strength of these validations.

Concerns about rule-based initialization: The rule-based construction of proxy cognitive structures in Stage I relies on the same interaction logs used for downstream tasks. This raised concerns about potential bias, limited independence of the learned representations, and the heuristic nature of the design.

Insufficient methodological analysis: The necessity of using diffusion models, as well as analyses of diffusion depth, threshold sensitivity, and comparisons with alternative graph generation methods, were considered insufficient.

Reproducibility and practicality issues: Questions remained regarding computational cost, scalability, and reproducibility, particularly due to the lack of publicly available code at the time of review.

**Reviewer Concerns:**

Unclear novelty: The rebuttal clarified that the main contribution is explicit cognitive structure graph generation, distinct from prior diffusion-based KT/CD methods that model latent states, and expanded the related work discussion. However, whether this distinction is sufficiently novel remains debatable.

Limited validation of cognitive structures: The rebuttal added human expert annotations, graph-level metrics, and comparisons with LLM-generated and random structures, which addressed the lack of direct evaluation. Still, the absence of an established ground truth for cognitive structures remains unresolved.

Concerns about rule-based initialization: The rebuttal explained that the rule-based proxy structures are only used for pretraining and can be corrected during the RL stage, and argued against label leakage. Nonetheless, concerns about heuristic bias and reliance on the same interaction logs persist.

Insufficient methodological analysis: The rebuttal justified the use of diffusion models with references to prior graph diffusion work, clarified notation, and added limited cost analysis. However, comparisons with alternative graph generators and deeper sensitivity analyses remain missing.

Reproducibility and practicality issues: The rebuttal added experiments on a larger dataset and inference-time comparisons, partially addressing scalability concerns. Full reproducibility remains outstanding due to unavailable code.

**Reviewer Scores:**

Overall, the rebuttal addressed many concrete technical and clarification-related concerns, particularly by adding empirical validations, scalability experiments, and cost analyses.
Reviewers who were initially positive would likely maintain their scores, as their main concerns were largely resolved.
Borderline reviewers might gain some confidence due to the additional experiments and clarifications, but would likely remain cautious because core issues such as the strength of the novelty claim and the difficulty of validating cognitive structures are judgment-based.
In contrast, the most critical reviewer would be unlikely to substantially increase their score, since their fundamental concerns about novelty, methodological necessity, and heuristic design were not fully resolved.

---

### Decision · Program_Chairs · 2026-01-26

Reject